# Microservices at Your Service:
# Bridging the Gap between NLP Research and Industry

**Tiina Lindh-Knuutila[1], Hrafn Loftsson[2], Pedro Alonso Doval[3],**
**Sebastian Andersson[1], Bjarni Barkarson[2], Héctor Cerezo-Costas[3],**
**Jón Guðnason[2], Jökull Snær Gylfason[2]**
**Jarmo Hemminki[1], Heiki-Jaan Kaalep[4]**

[1] Lingsoft, Turku, Finland
[2] Reykjavik University, Reykjavik, Iceland
[3] Gradiant, Pontevedra, Spain
[4] University of Tartu, Tartu, Estonia

{tiina.lindh-knuutila, sebastian.andersson, jarmo.hemminki}@lingsoft.fi
{hrafn, bjarnibar, jg, jokullg}@ru.is
{palonso, hcerezo}@gradient.org
{heiki-jaan.kaalep}@ut.ee

## Abstract

This paper describes a collaborative European project whose aim was to gather open source Natural Language Processing (NLP) tools and make them accessible as running services and easy to try out in the European Language Grid (ELG). The motivation of the project was to increase accessibility for more European languages and make it easier for developers to use the underlying tools in their own applications. The project resulted in the containerization of 60 existing NLP tools for 16 languages, all of which are now currently running as easily testable services in the ELG platform.

## 1 Introduction

Universities and other research institutes in Europe, and sometimes companies, are nowadays often publishing open source Natural Language Processing (NLP) software on various platforms, primarily GitHub. This software is often associated with research papers and, in the best case, also linked to other sharing platforms, such as CLARIN[1] or META-SHARE[2]. GitHub is, however, often the only place in which the tools are available. If a user finds a tool with a suitable license, it may still be difficult to determine if the tool works as intended. The threshold for trying out these NLP tools can also be high due to the reliance on various dependencies that may not be compatible with other desired tools or the tools are simply not up to date. Reproducibility of results is important in NLP but currently many results cannot be reproduced, even if the code is available. For example, Wieling et al. (2018) were only able to reproduce the same results in 1 out of 10 experiments.

In this paper, we describe a collaborative European project, *Microservices at Your Service: Bridging the Gap between NLP Research and Industry*[3] (hereafter simply referred to as the *Microservices* project), carried out by four partners: Lingsoft, a private company from Finland, University of Tartu from Estonia, Reykjavik University from Iceland, and Gradiant, a non-profit organisation from Spain. The main aim of the project was to increase accessibility of NLP tools for more European languages by:

- Making the tools available as running services in the European Language Grid[4] (ELG), and, additionally, registering them in ELRC-SHARE[5] for higher visibility and reach.

- Providing, for each tool, a tested container image which takes care of any dependencies and provides a logical handling of the data inputs and outputs, should the users want to use the container in their own computing environment.

---

[1] http://clarin.eu/
[2] http://www.meta-share.org/
[3] https://www.lingsoft.fi/en/microservices-at-your-service-bridging-gap-between-nlp-research-and-industry
[4] https://live.european-language-grid.eu/
[5] https://elrc-share.eu/

- Providing training and dissemination in the form of recorded workshops about the containerization of the tools, uploading the tools to the ELG.

- Finally, showcasing how the tools can be integrated for different purposes.

The project has deployed services in the ELG for 16 languages (see Section 4). For many languages, there is a distinct lack of resources in the current academic NLP research (Maria Giagkou, 2022). Highlighting the efforts made for low-resource languages is paramount to foster the development and usage of these resources by both the academic community and the industry, and our project targeted several of these low-resource languages.

Each underlying open source tool was implemented as a microservice (see Section 2.1) using Docker (see Section 3.3) for containerization. This allows developers, who need functionality from the various tools, to design their NLP applications as a collection of loosely coupled running services, as opposed to building the application using sources from various Github repositories, which, notably, may be written using various programming languages and depend on various external libraries.

In total, our project has resulted in the containerization of 60 existing NLP tools, all of which are currently running as services accessible through the ELG.

## 2   Background

Nowadays, software is often distributed to the end users via the Internet, rather than having the users install the software on their local machines. This method of distribution is called software-as-a-service or SaaS. Many large commercial organisations offer cloud platforms for distributing software, e.g. AI and NLP as SaaS, to the end users, and on some platforms it is possible for other organisations than the platform provider to upload their own tools for further distribution.

In this section, we provide the reader with basic information on the concept of microservices, the ELG cloud platform, and ELRC-SHARE.

### 2.1   Microservices

The microservice architectural style for software development has been defined as "[..] an approach to developing a single application as a suite of small services, each running in its own process and communicating with lightweight mechanisms, often an HTTP resource API" (Lewis and Fowler, 2014).

One of the advantages of microservices is that they can be updated without the need of redeploying the application that uses them. Another advantage is that different services can be implemented in different programming languages. In the contrasting monolithic architectural style, an application is built as a single executable unit (often using a single programming language). Any changes to the functionality demand building and deploying a new version of the application.

According to Francesco et al. (2017), "[m]icroservice architectures are particularly suitable for cloud infrastructures, as they greatly benefit from the elasticity and rapid provisioning of resources."

### 2.2   European Language Grid

The ELG is a scalable cloud platform, which hosts tools, data sets, and records of Language Technology (LT) projects and LT providers in official 24 EU languages and many additional ones. The goal of the ELG is to become the primary platform for LT, including NLP and speech technologies, in Europe. An important part of the purpose of ELG is to support digital language equality, "i.e., to create a situation in which all languages are supported through technologies equally well" (Rehm et al., 2021). Additionally, there is a growing movement to ensure that all relevant services can be offered by European providers to improve EU-wide digital sovereignty (European Parliament et al., 2023). Currently, most European cloud services are provided by non-European providers (Synergy Research Group, 2022).

The ELG platform is growing continuously and they foresee a need to evolve in the following areas: hardware capacity and cost distribution, hardware acceleration (for example, there is no GPU support yet), integration and deployment support, and workflow support (Kintzel et al., 2023).

ELG provides resources for developers to easily integrate a service: A (micro)service running in the ELG is wrapped with the ELG LT Service API and packaged in a Docker container. Both of these steps are carried out by the developer of the service. Thereafter, the container is integrated into

the ELG: It can either be called through the API or tested using a web UI. All APIs are https-based and use JSON as the primary data representation format. For easy creation of an application for an ELG-compatible service, Java- and Python-based libraries are available (Galanis et al., 2023).

For a user looking for potential tools, the ELG platform provides a faceted search functionality, allowing search by resource type such as corpus, tool, functionality, availability as an ELG-compatible service, data type, language, and license in a simple manner. The submissions to the service are also validated, which should improve the findability compared to a platform without such validation process.

## 2.3 ELRC-SHARE

ELRC-SHARE is a repository, maintained by the European Language Resource Coordination (ELRC)[6], for documenting, storing and accessing language data and tools in all EU languages, Norwegian Bokmål, Norwegian Nynorsk, and Icelandic. The original intent of the repository was to obtain and store data and tools that contribute to the European Commission's automated `eTranslation` platform[7], but the scope has broadened to include other LT tools as well. Approximately 80% of the language resources are freely usable outside ELRC (Marra et al., 2022).

## 3 Project Execution

Our two year project started in March 2021. The goal of the project (described in Section 1) included several stages. In the first stage, we sought out open source tools that might be of potential interest. We prioritized those that are actively maintained or developed. This was carried out both by bottom-up search on the software sharing platforms (primarily GitHub), and by contacting research institutions in the targeted regions. In parallel, we also collected standard or available test data sets for the tools. This initial phase was followed by testing the set of collected tools on the existing test data. If many tools existed for the same task, a selection was made based on metrics performance and language coverage. After all tools were tested and selected, we started containerizing the tools and expose a web service

API for each of them on the ELG. Finally, we stored metadata information of each tool in ELRC-SHARE. Our dissemination activities ran parallel to making the tools available: We held workshops on different themes of the project, ranging from dockerization of the tools to demonstrating their functionality and use case integration.

### 3.1 Searching for tools

The search for tools was not primarily guided by pre-specified project goals or use cases, but rather guided by the subjective explorative interests of the individual partners.

At the start of the project, there was an initial assumption made that university labs or individual programmers were storing interesting and useful tools on local disks. These tools could then be made public via the project. However, the reality was different: source code was always in GitHub[8] or GitLab[9]. The focus therefore quickly shifted to verifying that the found tools were functioning well.

To find interesting tools, we sent emails to university contacts, browsed university web repositories and arXiv, did online searches with relevant keywords (e.g. 'speech recognition', 'parsing', or 'named entity recognition') and looked up conference proceedings and journal articles for interesting repositories. Then, we went through each promising repository to see first if all the relevant parts for running the tool were available. This was followed by an initial compilation of the tool and ensuring that we obtained the same or at least similar results as the original authors, if the test data was available. If not, we gathered examples to ensure the test results seemed reasonable.

### 3.2 Testing and documenting

To make a third-party tool available for the wider public involves providing documentation, which minimally describes the following: a) What the purpose of the tool is; b) how to run the tool; c) specification of the tool input and output formats and error handling; d) the original authors of the tool; and e) what kind of a licence or terms of use the tool has.

Often these points have already been addressed by the authors of the tool, although the amount of details varied. We sometimes had to fill in missing

---

[6]`https://www.lr-coordination.eu/`
[7]`https://webgate.ec.europa.eu/etransl ation/public/welcome.html`

[8]`http://github.com/`
[9]`http://gitlab.com/`

information (most notably the licence) and come up with our own wording about the purpose and place of the tool in the ecosystem of the LT field of the particular language.

While creating the documentation for the microservices tools it was noticed that some tools with similar functionality had differing output types without an explicit reason why. Such differences can of course be justified, but can also indicate that some standardisation in a field might benefit interoperability. This was especially notable for morpho-syntactic categories for Estonian and University of Tartu set up a designated webpage[10] for facilitating comparison between these identified systems.

### 3.3 Dockerization

We used Docker[11] for developing, distributing and running the NLP tools (in the ELG). Docker has in recent years been established as a convenient solution for making it easier to create, deploy, and run applications by using containers. Containers allow developers to package up an application with all requirements, such as libraries and other dependencies, and distribute it as a single stand-alone package. Docker is a good option for a platform independent solution for making NLP tools available for both researchers and software developers.

Each of the selected NLP tools was dockerized by building a container with the tool itself along with an http API that gives people/programs outside the container access to the tool. All of the images for our tools are shared in the Docker Hub[12], world's largest library for container images. The difficulty of dockerizing a given NLP tools was dependent on how easy it was to give the API in the container access to the tool. Once the API was able to receive output from the NLP tool, all that was left was to make sure that the output from the API was in accordance to the ELG specification.

For each service integrated to the ELG, we also provided metadata, which contains a link to the code repository of the underlying tool.

### 4 The NLP Tools

In our project, the focus was on tools for the Nordic/Scandinavian languages, the Baltic languages, and the Iberian languages, simply because

of the partners' geographical locations and local interests.

We dockerized 60 existing NLP tools, in 16 languages: Catalan: 2; English: 2; Estonian: 11; Faroese: 1; Finnish: 4; Galician: 1; Basque: 1; Icelandic: 11; Komi: 1; Latvian: 3; Lithuanian: 2; Northern Sami: 2; Norwegian: 1; Portuguese: 6; Spanish: 5; and Swedish: 3. Additionally, we provided four multilingual tools. Whilst the majority these tools come from European institutions, the project also made available relevant results from South American countries (Brazil, Chile and Uruguay).

The list of dockerized tools is available at the project website. The NLP tools are very diverse, covering from low level (e.g. PoS taggers, morphological analyzers, NERs and parsers) to high level applications (e.g. question answering (QA) and audio processing), as well as others with niche results (detection of false friends and text generation of proverbs given a short text)

### 5 Getting the Tools into Use

There is a risk that new tools made for low-resource languages might not be known by the community. A tool might be created as a one time release for an academic publication, or it might not have gathered the attention needed for a continued development. For the purpose of both stimulating researchers to share their tools and promote the tools we made available, we held three types of workshops: First we had an early awareness workshop, in which we provided hands-on guidance on how to release available tools as Docker images. During the second year, we held two workshops focusing on how to make tools available in the ELG platform. Finally, at the end of the project, we held workshops which summarized our work and demonstrated how the tools we provided can be integrated into LT applications. All workshops are made available on the project webpage.

In what follows, we describe some of these pilot integration cases. In each of these cases, it was easy to "plug in" a container with a well defined API, and then handle the input and output in the process pipeline.

A language identification (LID) tool was utilized in two different cases. In a translation process, we utilized it to make sure the training data for a neural machine translation (NMT) model was actually in the correct language. The original

---

[10]https://cl.ut.ee/ressursid/morfo-systeemid/
[11]https://www.docker.com/
[12]https://hub.docker.com/

texts contained sentences in other languages, causing an in-production NMT model to occasionally produce English instead of Swedish translations. The previous LID tool had a 98.9 % precision and 96.4% recall for Swedish, whereas the new tool, HeLI OTS (Jauhiainen et al., 2022), had a 99.9 % precision and 99.6% recall. When there are hundreds of millions of words in the training material, one percentage point yields millions of words tagged in wrong language. The new LID tool alleviated this problem to a sufficient extent.

This LID tool was also found useful in an online library platform[13], where publishers provide large amounts of e-books. Sometimes the metadata provided by the publisher does not match the language of the actual e-book, yielding erroneous behavior, for example, in screen readers.

At the online library platform, we also piloted aligning audio books and e-books, to allow seamless switching from text to audio and back, using an audio alignment tool. This tool was not designed for this kind of task originally, but, nevertheless, it allowed testing potential new features for the platform. Furthermore, we also tested NER and linking to ontologies to further improve the findability within an e-book or audio book.

## 6 Limitations

In the previous sections, we have argued that it can be beneficial to dockerize NLP tools for the purpose of making them accessible as running microservices. However, this approach can have some practical limitations.

First, changes to a tool do not automatically become available in the dockerized version. Thus, the running microservice in the ELG might become outdated. However, if the developer of the underlying tool is keen on making the newest version running as a microservice, the developer can easily build the docker image again (the code for building it is open source) and then ask ELG to pull the new image from the associated docker hub. Most of that process can also be automated.

Second, due to resource constraints, ELG services are not guaranteed to be constantly running. If a user calls the API of a service, which is not running, the user will probably experience considerable initial delay (associated with the first API call) before the requested service has started.

With regard to both of the above mentioned limitations, it is worth noting that anyone can use a given docker image to expose an API for the underlying tool on some web server. In other words, ELG is not the only option for providing access to a running service.

## 7 Conclusion

In this paper, we have described a collaborative project which succeeded in making 60 NLP tools covering a total of 16 languages available as microservices in the ELG platform. We also described the microservice principles and the European platforms that record or host these microservices, and the steps to get the tools into these platforms.

We recommend that researchers continue this work by providing their tools as Docker images and as compatible services in the ELG platform. This requires just a little more effort from the researchers, but substantially lowers the threshold for testing the tool for new researchers/developers. Hence, lowering the threshold for integrating the tool in new services and raising the potential impact of the initial research.

## Acknowledgments

This project has been co-financed by the Connecting Europe Facility of the European Union, project number 2020-EU-IA-0046.

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
