# OpenReview forum: "Microservices at Your Service: Bridging the Gap between NLP Research and Industry"
_NoDaLiDa/2023/Conference — NoDaLiDa 2023_

### Official Review · Reviewer_9cru · 2023-03-06
**Nice description of the project but lacking clear statements of findings**

**Rating:** 4
**Confidence:** 4

**Review:**

The paper describes an EU-funded project that had the goal to integrate existing NLP tools as in the European Language Grid. To do so, the various components have been implemented in Docker containers which are easy to deploy and, as far as I understand from the paper, also easy to be offered as software-as-a-service. The paper does a good job in explaining the advantages of providing NLP tools as SAAS and transparently explains the required work steps to do so. It provides some insight in the observations that the researchers made during this project.

What I am missing in the paper, however, is a focus on the main findings in the project. The impact for future work remains unclear. This is mostly an effect of this paper not stating clearly what the target reader or user is.

Questions for which I'd like to see answers in this paper are:

* What are the main findings from the integration effort? What were the main challenges and how could future NLP tool developers support the integration in the ELG?
* How does the described project support the integration of future tools?
* The advantages of such integration are quite clear (easier usability), but what are disadvantages? Does such integration come with a slower runtime? Or with a limitation to particular computer environments?
* What are best-practives for future integrations in the ELG and what are the advantages of following them?

Some more detailed/smaller comments:

* I'd really like if the abstract mentioned the main findings of this project. What do we know now that we could not know before this project took place?

* Why is Section 3.1. called "Hunting"? This suggests that the projects try to "escape", but I am pretty sure that most tool developers would be very happy to have their project integrated.

* The research process of finding systems (lines 254ff) is not very transparent. You mention to use "relevant keywords" but it remains unclear what these are.

* What would have been alternatives to using docker?

* What was the inclusion/exclusion criterium for tools regarding software properties and/or target NLP tasks?

In summary, the paper is promising, but needs to be more detailed regarding the approach and main findings. In its current state it is of limited usefulness for potential users, but I believe that the authors could quite easily make it a very useful contribution. I believe that another review round would be beneficial for this paper.

**Paper Type:**

Demo

---

### Official Review · Reviewer_q2nb · 2023-03-09
**Project report with significant practical results**

**Rating:** 7
**Confidence:** 5

**Review:**

This paper presents the outcome of a project linked to the European Language Grid. In this project, a large number of NLP systems have been collected, wrapped in microservices with an API, dockerized, and uploaded into the platform, which will provide long-term maintenance and support for them. This has important impact e.g. on reproducibility of the results of these systems, but it also contributes to the language equality idea, since the language covered are many.
Having worked on importing NLP systems into ELG myself, I am confident that a great deal of work must have been put into this project.
A few questions are underspecified in the paper. Mostly, it is not clear to this reader how and why the systems were selected. Were there specific criteria? Was the selection based on the languages still missing from the platform?
Moreover, a bit more information on what metadata were provided along with the systems would be beneficial, since this is an important feature on ELG, e.g. for discoverability of its services.
I am suggesting a demo presentation because it may be a good way of showcasing the project results, in my experience.

**Paper Type:**

Demo

---

### Official Review · Reviewer_K9GV · 2023-03-12
**title of the paper is not given - "will be provided in the final version"**

**Rating:** 4
**Confidence:** 5

**Review:**

The paper raises several valid points:
A. there is a growing need on the part of developers — academic and commercial — to integrate pre-existing applications / tools, or tools developed and shared by others into their own research or applications.
B. finding the correct tools that would be suitable / helpful for my application is not an easy task, requiring searching through various sources.

These are unquestionably valid concerns.

The paper offers a project that sought out 60 LT tools, dockerized them, and made them available through the ELG (European language grid).

The benefit of pre-dockerization of these 60 hand-picked LTs which were then integrated — is not made clear in the paper.

Outside of LT, the Github approach is accepted worldwide by everyone in computer science / data science, and recently far beyond. It is not made clear why yet another effort needs to be expended on standardizing and sharing of LT tools. We already have a variety of platforms for gathering and maintaining LT tools: including Github and its variants, ELRA/ELDA, LDC, the CLARIN infrastructure, etc.

One objection to the dockerization-based approach suggested in the paper is — especially if the tool is actively developed and maintained  — any changes made to the tool are not available in the dockerized version offered by this project.  The versions in Github are expressly guaranteed to be the latest, the most stable, or both.

Another objection is that the "real" challenges — for example B above — are in fact not addressed at all: the authors undertake an ad hoc approach to finding some tools (section 3.1), as all of us do — offering no solutions to this very real problem. Novel methods / workflows to address B would add real value; picking 60 tools to dockerize does not.

It would have been interesting and valuable if the paper were to address the difficulties in the original challenged — A and B above.  Section 5 promises case-studies or specific use-cases that would confirm the usefulness of the proposed approach.

The use-cases would have been more informative if they could present a real-world end-to-end use case, where the proposed resource actually makes a difference — enables one to do something that is not otherwise possible, or is otherwise very difficult to achieve. No such use-case is presented in Section 5.

Minor comments:

Insisting that two tools that perform a similar function must have the same output format (section 3.2, the case of Estonian) is not sensible for two reasons.  1. the authors of the tools may have specific reasons why they want the output to look a certain way; 2. even if such fundamental rationale is lacking, converting the output from format A to format B is an exercise for a beginning BA student, requiring a minimal investment of time and resources. Therefore it does not appear to be a worthwhile investment of resources

In a few places there are typos, they should be fixed: line 314, etc.


**Paper Type:**

Short paper

---

### Decision · Program_Chairs · 2023-03-17

Accept